# New Mutant Amaranth Varieties as a Potential Source of Biologically Active Substances

**DOI:** 10.3390/antiox10111705

**Published:** 2021-10-27

**Authors:** Jozef Fejér, Ivan Kron, Adriana Eliašová, Daniela Gruľová, Alena Gajdošová, Veronika Lancíková, Andrea Hricová

**Affiliations:** 1Department of Ecology, Faculty of Humanities and Natural Sciences, University of Prešov, 17. Novembra St. 1, 080 01 Prešov, Slovakia; jozef.fejer@unipo.sk (J.F.); adriana.eliasova@unipo.sk (A.E.); 2JNT Ltd., 040 01 Košice, Slovakia; kron.ivan@gmail.com; 3Institute of Plant Genetics and Biotechnology, Slovak Academy of Sciences, Plant Science and Biodiversity Center, Akademická 2, P.O. Box 39A, 950 07 Nitra, Slovakia; alena.gajdosova@savba.sk (A.G.); veronika.lancikova@savba.sk (V.L.); andrea.hricova@savba.sk (A.H.)

**Keywords:** *Amaranthus cruentus* L., *Amaranthus hypochondriacus* × *Amaranthus hybridus* L., flavonoids, free radicals, total polyphenols, varieties ‘Pribina’ and ‘Zobor’

## Abstract

Amaranth species represent a diverse group of plants. Many of them are a rich source of secondary metabolites with many positive biological effects. Total phenolic, total flavonoid and rutin content, antioxidant activity against superoxide and hydroxyl radicals, FRAP (Ferric-reducing ability of plasma) assay and DPPH (2,2-Diphenyl-1-picrylhydrazyl) radical scavenging assay were determined in ethanol extracts of dried leaves of the new Slovak amaranth varieties ‘Pribina’ and ‘Zobor’. The amount of total phenolic substances (‘Pribina’ GAE 38.3 mg.g^−1^ DM and ‘Zobor’ GAE 26.1 mg.g^−1^ DM), content of total flavonoids (‘Pribina’ QE 26.5 mg.g^−1^ DM and ‘Zobor’ QE 20.3 mg.g^−1^ DM) and rutin (‘Pribina’ 50.8 mg.g^−1^ DM and ‘Zobor’ 15.2 mg.g^−1^ DM) were higher in the variety ‘Pribina’, compared to the variety ‘Zobor’. A statistically higher antioxidant activity against superoxide radical (1.63%·mg^−1^g^−1^ DM), hydroxyl radical (3.20%.mg^−1^g^−1^ DM), FRAP assay (292.80 µmol.L^−1^·mg^−1^.g^−1^ DM) and DPPH (54.2 ± 1.78 µg.mL^−1^ DM) were detected in the ‘Pribina’ variety. Antiradical and antioxidant activities of both extracts showed high positive correlations in relation to the content of total phenolic substances, total flavonoids and rutin. Amaranth is an undemanding crop on specific environmental conditions and is resistant to abiotic and biotic stress.

## 1. Introduction

Pseudo-cereals like amaranth (*Amaranthus* spp.) with a large number of variable species can markedly contribute to the protection of the agricultural environment, to sustaining its biodiversity, to the global food production as well as to the preparation of healthy foods and food additives. As a grain and vegetable crop, amaranth is able to grow under wide climatic conditions [1]. Naturally occurring species of amaranth and their hybrids have numerous possibilities for utilization in Slovakia [2]. As the result of inducing mutations by irradiation of *Amaranthus cruentus* L. genotype ‘Ficha’, in the year 2013, in the Slovak Republic, the variety ‘Pribina’ was created and registered [3]. The breeding effort for the interspecific hybrid *Amaranthus hypochondriacus* × *Amaranthus hybridus* L. K-433 resulted in the release of the variety ‘Zobor’ in the year 2016, by the same method [4]. A high-weight seed was achieved by mutagenesis, which was genetically fixed by selection. These cultivars are intended for seed production [5]. Due to the nutritional value and potential health benefits of amaranth seeds, the plant has received much attention in recent years. Amaranth is rich in macronutrients (proteins, carbohydrates and fats) and micronutrients, including vitamins and minerals [6]. Past studies of amaranth antioxidant activity were mostly performed with seeds, sprouts, by-products [7,8,9,10] and leaves [11,12,13,14,15], or even with amaranth oil [16].

Reactive oxygen species are formed in living organisms during metabolic processes. These are the problem of many chronic diseases caused by oxidative stress. Therefore, antioxidants in foods and pharmaceuticals are important components of the body’s defenses. A large number of aromatic, spicy, medicinal and other plants exhibit antioxidant properties. Many studies confirm the antioxidant activity of extracts made from different parts of *Amaranthus* spp. [17,18]. Natural sources of antioxidants are primary plant phenolic compounds that may occur in all parts of plants, such as the fruits, vegetables, nuts, seeds, leaves, roots and bark [19]. Radicals and reactive oxygen species (ROS), which include free radicals such as superoxide anion radicals (O_2_^•−^), hydroxyl radicals (HO^•^) and non-free radical species such as H_2_O_2_ and singlet oxygen (^1^O_2_), are various forms of activated oxygen. These molecules exacerbate cellular injury and the aging process [20]. It is well established that organelles such as chloroplasts, mitochondria or peroxisomes with a highly oxidizing metabolic activity or with intense rates of electron flow are a major source of ROS in plant cells. The appearance of O_2_ in the atmosphere enabled respiratory metabolism and efficient energy generation systems, which use O_2_ as the final electron acceptor, and lead to the formation of ROS in cells [21]. Although atmospheric oxygen is relatively non-reactive, it can give rise to ROS, which includes O_2_^•−^, H_2_O_2_, HO^•^ and ^1^O_2_ [22]. HO^•^ is among the most highly reactive ROS known [23]. The generation of hydroxyl radicals is crucial for the irreversible damage inflicted by oxidative stress. This generation mainly proceeds via the Fenton reaction [24]: H_2_O_2_ + Fe^2+^ → Fe^3+^ + HO^−^ + HO^•^.

The test measuring the ferric reducing ability of plasma, the FRAP assay, is presented as a method for assessing ‘‘antioxidant power’’. Ferric to ferrous ion reduction at low pH causes a colored complex to form ferrous-tripyridyltriazine [25]. A biological antioxidant has been defined as any substance that, when present at low concentrations compared to those of an oxidizable substrate, significantly delays or prevents oxidation of that substrate [26].

The aim of our research was to determine the antiradical, and antioxidant activity of ethanol extracts from the leaves of two new Slovak varieties of amaranth, ‘Pribina’ and ‘Zobor’, using four methods with different mechanisms of action. At the same time, we determined the content of total phenolic substances, total flavonoids and the major flavone rutin with the assessment of their effect on the total antioxidant activity and scavenging of free radicals.

## 2. Materials and Methods

### 2.1. Plant Material and Extract Preparation

The plant material, amaranth leaves from *Amaranthus cruentus* L. variety ‘Pribina’ and *Amaranthus hypochondriacus* × *Amaranthus hybridus* L. variety ‘Zobor’, were grown on an experimental field belonging to the University of Prešov (48°59.382′ N, 21°13.576′ E) at 253 m above sea level. Leaves were harvested at the butonisation stage and naturally dried. The dried leaves were crushed in a porcelain mortar with a pestle. Then, 10 g of powder were dissolved in 100 mL of 70% ethanol. The extraction was carried out for 72 h at room temperature. Obtained extracts were filtered over filter KA 1-M (very fast). The dry matter (DM) content was determined in the filtrates. All chemicals used were of the highest quality. Double distilled water (DDW) was used for the preparation of solutions. Absorbance of solutions in various test assays was determined with a Shimadzu type UV-1800 spectrophotometer (Shimadzu, Japan).

### 2.2. Superoxide Anion Radical Scavenging Activity

The assay was provided based on the published method [27]. The details can also be found in our previous publications [28,29]. The percentage of inhibition (Superoxide [%] in Table 1) was calculated. All determinations were performed at least four times.

### 2.3. Hydroxyl Radical Scavenging Activity

The assay for scavenging hydroxyl radicals based on deoxyribose was inspired by the published method [30] with small modifications [28,29]. The percentage of inhibition (Hydroxyl [%] in Table 1) was calculated. All determinations were performed at least four times.

### 2.4. Ferric Reducing Ability of Plasma (FRAP) Assay

The working FRAP reagents and other reagents were prepared according to the published method [25]. The only change was in the increase of hydrochloric acid concentration to 50 mmoL.L^−1^ for dissolving of 10 mmoL.L^−1^ TPTZ (2,4,6-tripyridyl-s-triazine) [31]. The activity was expressed in Gallic acid equivalents GAE (FRAP [µmol.L^−1^] in Table 1). All determinations were performed at least four times.

### 2.5. DPPH (2,2-Diphenyl-1-picrylhydrazyl) Radical Scavenging Assay

Free radical scavenging ability of the extracts was evaluated by DPPH radical scavenging assay according to a described procedure with slight modifications [32]. A stock solution of DPPH (0.06 mM) was prepared in methanol. DPPH stock solution (0.9 mL) was well-mixed with 0.1 mL plant extract of various concentrations (i.e., original extract diluted with methanol) and left in the dark at room temperature for 30 min. The absorbance of the reaction mixture was measured at 517 nm against methanol as a blank with a Shimadzu UV-1800 spectrophotometer. Ascorbic acid (Sigma-Aldrich, Saint Louis, MO, USA) was used as a reference antioxidant. Rutin (Acros Organics, Geel, Belgium), the major flavone found in the extracts, was also assayed. Percentage DPPH radical scavenging activity was calculated by the equation: % DPPH = [(A_0_ − A_S_)/A_0_] × 100, where A_0_ is the absorbance of the control (reaction mixture without extract/reference antioxidant), and A_S_ is the absorbance of the test sample (reaction mixture with extract/reference antioxidant). Half maximal inhibitory concentration (IC_50_) value, i.e., the concentration of the plant extract/reference antioxidant that could scavenge 50% of DPPH radical, was calculated using the graph, in which % DPPH was plotted against concentration (μg DW.mL^−1^). All experiments were repeated four times at each concentration. All solutions were used on the day of preparation.

### 2.6. Determination of Total Phenolic Content

The total contents of phenolic compounds in the extracts were determined by using the Folin-Ciocalteu phenol reagent (FCR, Merck) according to the described method with slight modifications [33]. Prior to the assay, the extracts were diluted with 70% ethanol in the ratio of 1:4 (*v/v*). Each diluted extract (0.1 mL) was sequentially well-mixed with 0.2 mL of FCR, 2 mL of DDW and 1 mL of Na_2_CO_3_ saturated solution (20%, *w/v*, in water). After incubation in the dark at room temperature for 90 min, the absorbance of the reaction mixture was determined at 765 nm with a Shimadzu UV-1800 spectrophotometer. The blank sample was prepared in a similar way by replacing the extract with the same volume of 70% ethanol. The phenolic content was calculated as Gallic acid equivalents (GAE) per mL of extract on the basis of a standard curve of Gallic acid (Merck). All determinations were carried out four times for each extract. All solutions were used on the day of preparation.

### 2.7. Determination of Total Flavonoid Content

The total contents of flavonoids in the extracts were determined using the aluminum chloride colorimetric method according to a reported procedure with slight modifications [34]. Prior to the assay, the extracts were diluted with 70% ethanol in the ratio of 1:4 (*v/v*). Each diluted extract (0.2 mL) was sequentially well-mixed with 1.8 mL of 70% ethanol, 0.1 mL of AlCl_2_ solution (10%, *w/v*, in methanol), 0.1 mL of 1M CH_3_COOK and 2.8 mL of DDW. After incubation at room temperature for 30 min, the absorbance of the reaction mixture was measured at 415 nm with a Shimadzu UV-1800 spectrophotometer. The blank was prepared in a similar way by replacing AlCl_2_ solution with the same volume of DDW. The flavonoid content was calculated as quercetin equivalents (QE) per mL of extract on the basis of a standard curve of quercetin (Sigma-Aldrich). All determinations were carried out four times for each extract. All solutions were used on the day of preparation.

### 2.8. HPLC-DAD Analysis of Rutin (Quercetin-3-O-rutinoside)

Plant extracts were filtered through a 0.45 μm nylon membrane syringe filter (Puradisc 13, Whatman, papírna Perštejn, Czech Republic). The filtrates were analyzed by gradient reverse-phase HPLC on a Dionex UltiMate 3000 Quarternary Analytical System with diode array detector and Dionex Acclaim 120 C18 column (5 μm, 250 × 4.6 mm), maintained at 25 °C. Mobile phase A (water for HPLC with 0.1% formic acid, *v/v*) and mobile phase B (acetonitrile for HPLC, gradient grade with 0.1% formic acid, *v/v*) were derived from the column at 1.0 mL.min^−1^ according to a gradient program as follows: 0 min, 2% B; 0–33 min, 2–30% B; 33–37 min, 30–100% B; 37–42 min, 100% B; 42–45 min, 100–2% B; 45–50 min, 2% B. The peak of rutin was identified based on the retention time, and UV-VIS spectra measurements (200–500 nm) were carried out during the analysis by comparison of the corresponding standard compound (ACROS Organics). For quantitative evaluation, the peak area values obtained at a wavelength of 350 nm were used. The calibration line consisted of four calibration points, each point being the result of three injections of standard solution. It showed good linearity over the entire range of concentrations tested (125–1000 mg.mL^−1^, R^2^ > 0.999).

### 2.9. Statistical Analysis

The statistical software was used Statgraphics 5.0 to perform a multifactorial analysis of variance (MANOVA), and the Tukey test was used post-hoc where there were significant differences between the means. A 95% confidence interval was used for the statistical analysis.

## 3. Results

Results of superoxide anion radical scavenging activity, hydroxyl radical scavenging activity, FRAP (ferric reducing ability of plasma) assay and DPPH (2,2 Diphenyl 1 picrylhydrazyl) are shown in Table 1.

As shown in Table 1, the statistical analysis confirmed that ethanol extract of ‘Pribina’ contained more dry matter than extracts from ‘Zobor’. All methods showed higher antioxidant activity of the ethanol leaf extract of the amaranth variety ‘Pribina’, compared to the extract of the variety ‘Zobor’. These differences were highly statistically significant. Sodium salicylate was used as a reference standard for superoxide anion radical scavenging activity, which showed a pro-oxidative activity of −14.08 ± 3.97% (concentration of SA 1.0 mmol.L^−1^) compared to the extracts. Gallic acid was used as a standard in the hydroxyl radical evaluation and in the FRAP method. In a series of hydroxyl radical scavenging activities, Gallic acid showed 48.28 ± 3.72% (concentration of GAE 1.0 mmol.L^−1^) antioxidant activity. The FRAP method showed a Gallic acid antioxidant effect of 5276.9 ± 182.0 µmol.L^−1^ (concentration of GAE 0.1 mmol.L^−1^). In the case of the DPPH method, the IC_50_ of ascorbic acid was 5.5 ± 0.02 µg DM.mL^−1^, which was significantly higher than in the extracts of both varieties. In Table 2 we can see the content of total phenolic, flavonoids and rutin.

The content of total phenols, flavonoids and rutin were higher in the leaf extract of the ‘Pribina’ variety compared to the extract of the ‘Zobor’ variety. In all cases, the differences were highly statistically significant (Table 2). Determination of rutin was provided by the HPLC-DAD and is presented in the chromatogram (Figure 1).

The obtained values of all types of antioxidant activity and content of total phenols, flavonoids and rutin in extracts were recalculated on dry matter. We consider such values more precise for the evaluation of antioxidant activity because we can consider the amount of extracted DM and the number of phenolic substances in it [28,29]. Correlation analysis was performed from the recalculated values on DM. A highly significant correlation was found between individual types of antioxidant activity and the content of total phenols, flavonoids and rutin (Table 3).

Extract preparation and quercetin-3-O-rutinoside (rutin) evaluation procedures are described in the Materials and Methods section.

## 4. Discussion

Aqueous extracts of *A. cruentus* leaves showed antioxidant activity against the superoxide radical in the range from 1.0% to 46.9% [35]. The antioxidant activity of ethanol leaf extracts against the superoxide radical was measured as 1.08 ± 0.09%.g^−1^ DM (variety ‘Zobor’) and 1.63 ± 0.14%.g^−1^ DM (variety ‘Pribina’) in our research. We found a relatively high antioxidant activity against the hydroxyl radical. The ethanol extract of the ‘Pribina’ variety had 3.20 ± 0.09%.g^−1^ DM and the extract from the ‘Zobor’ variety showed 2.13 ± 0.14%.g^−1^ DM efficiency. The antioxidant effect of Gallic acid was 48.28 ± 3.72% (in concentration of GAE 1.0 mmol.L^−1^). The dose-dependent effect of hydroxyl radical scavenging activity for methanol, ethyl acetate and aqueous extracts from *A. lividus* plants has been reported [36]. At a concentration of 20 mg.mL^−1^, the methanol extract showed 92.8 ± 0.08% activity, the ethyl acetate extract had 91.7 ± 0.78% activity, and the aqueous extract had 60.0 ± 1.83% activity, in the abovementioned research. Gallic acid had 65.4 ± 0.85% activity, which is comparable to the aqueous extract. It has also been found that with increasing concentration of extract, the antioxidant activity against the hydroxyl radical increases in aqueous leaf extracts of *A. cruentus* [35]; it ranged from −2.7% to 68.9%. The substances present in the amaranth leaves of both varieties, ‘Pribina’ and ‘Zobor’, were more effective against the hydroxyl radical than they were against the superoxide, though in the latter case, it was due to the long incubation time (40 min). A more complicated electron reception and transfer mechanism can be like simple phenols. Several simple phenols revealed antioxidant activity changing to pro-oxidant activity with time [37,38]. It has also been reported that DPPH scavenging activities of the extracts *A. lividus* (dried stems with leaves and flowers), expressed as an IC_50_ value, exhibited the strongest antioxidant activity of the ethyl acetate extract 6.75 ± 0.08 mg.mL^−1^, followed by the methanol extract 24.8 ± 0.36 mg.mL^−1^ and the water extract 42.3 ± 0.86 mg.mL^−1^ [36]. High DPPH activity was found for *A. cruentus*; the IC_50_ was 0.294 ± 0.13 µg.mL^−1^ DM and 0.392 ± 0.07 µg.mL^−1^ DM for *A. hypochondriacus* [39]. Different DPPH activity IC_50_ depending on the solvent type was detected in a previous study [40]. Hydroacetonic extract had an activity of 75.6 ± 0.5 μg.mL^−1^, methanol extract 95.0 ± 4.0 μg.mL^−1^ DM and aqueous extract 330.0 ± 10.0 μg.mL^−1^ DM activity. Ethanol leaf extract of variety ‘Pribina’ showed antioxidant activity DPPH IC_50_ 54.2 ± 1.78 µg.mL^−1^ DM and of variety ‘Zobor’ 66.9 ± 0.74 µg.mL^−1^ DM. Antioxidant activity of fruit methanol extract of *A. cruentus* IC_50_ 38.48 ± 3.03 µg.mL^−1^ DM was presented [41]. Antioxidant activity using the FRAP method depended on the growth stage and was evaluated in methanol extracts of *A. caudatus* [42]. The antioxidant activity ranged from 469.0 ± 75 µmol Fe^2+^.g^−1^ of extract (growth stage budding) to 830.0 ± 27 µmol Fe^2+^.g^−1^ of extract (growth stage early vegetative). In our experiments, determined antioxidant activity was 286.5 ± 11.7 µmol.L^−1^.g^−1^ DM (‘Pribina’ variety) and 196.6 ± 8.9 µmol.L^−1^.g^−1^ DM (‘Zobor’ variety). The content of total phenols in the ethanol extract of *A. cruentus* leaves was determined in the amount of 34.93 ± 7.19 mg.g^−1^ DM and the content of total flavonoids 32.31 ± 7.77 mg.g^−1^ DM [39]. The extract of *A. hypochondriacus* leaves had a slightly lower content of total phenolic substances 28.05 ± 3.74 mg.g^−1^ DM as well as flavonoids 25.38 ± 2.03 mg.g^−1^ DM. The amount of total phenols (1460.0 µg.mL^−1^) and rutin content (302.0 µg.mL^−1^) in the ethanol leaf extracts of *A. cruentus* were observed in [43]. Another study determined the content of total phenolic in methanol extract of *A. cruentus* as 42.93 ± 1.12 mg.g^−1^ GAE and content of total flavonoids mg.g^−1^ CAE (Catechin equivalent) [44]. In hydroacetonic (HAE), methanol (ME) and aqueous extract (AE) from aerial parts of *A. cruentus*, the content of total phenolic 10.18 ± 0.60 mg.100 mg^−1^ GAE, 7.55 ± 1.18 mg.100 mg^−1^ GAE and 8.40 ± 2.69 mg.100 mg^−1^ GAE were found [40]. In contrast, a low content of phenolic substances in the extract from the aboveground parts of *A. lividus* plants was evaluated [36]. The amount of total phenolic compounds per gram of dry matter in the water extract was 1.55 ± 0.098 mg.g^−1^ DM, in the methanol extract 1.51 ± 0.13 mg.g^−1^ DM and the ethyl acetate extract had the lowest content of 0.46 ± 0.039 mg.g^−1^ DM. The content of rutin depending on the dose of fertilizer (2402.0 mg.g^−1^) and the date of harvest time (2279.0 mg.g^−1^) was determined in the aqueous leaf extract of *A. cruentus* [35]. In our experiments, we have found the amount of total phenols in the ‘Pribina’ variety 38.1 ± 12.01 mg.g^−1^ DM and in the ‘Zobor’ variety 26.1 ± 0.33 mg.g^−1^ DM. The total flavonoid content was 26.5 ± 0.07 mg.g^−1^ DM in ‘Pribina’ and 20.3 ± 0.26 mg.g^−1^ DM in ‘Zobor’. The mechanism of the antioxidant activity of phenolic acids is based on the provision of a hydrogen atom from an antioxidant molecule. Similarly, the antioxidant activity of flavonoids is manifested mainly in their ability to provide a hydrogen atom to other molecules, to bind metals to complexes to reduce α-tocopherol radicals and thus to regenerate α-tocopherol, as well as to scavenge singlet oxygen [45,46]. Our experiment confirmed the high correlation of individual methods of antioxidant activity with the content of total phenolic substances, total flavonoids and rutin (Table 3). Similar conclusions were also noted in [36]. A high correlation with DPPH activity (r = 0.992, 0.994 and 0.999 for water, methanol and ethyl acetate extracts, respectively) and hydroxyl radical scavenging activity (r = 0.973, 0.832 and 0.917 for water, methanol and ethyl acetate extracts, respectively) through the low content of phenolic substances in the extracts of the aboveground parts of *A. lividus* was found. Despite the inconsistent methods of processing amaranth plant material, some authors confirm similar conclusions regarding the antioxidant properties of amaranth extracts and their relation to the total phenol content. Positive correlations between FRAP and DPPH with total phenolic contents of methanol, hydrochloric acid and water mixture extractions of *A. cruentus* seeds and sprouts were determined [9]. A very high correlation (r = 0.908) between antioxidant activity (DPPH assay) and total phenols suggests that phenolic compounds are the major antioxidant components in the methanol extract of leaves of seven amaranth cultivars. Total phenols and antioxidant activity were greater in the leaves from plants grown under full sunlight without shading [12,13]. A direct relationship between FRAP activity and the phenol content of ethanol extract of leaves of *A. tricolor* was found [47]. However, higher phenolic content and lower antioxidant activity (DPPH assay) were observed in aqueous extracts of leaves of *A. tricolor* and *A. viridis*. The effect of pH (5.0, 7.2 and 9.0) and temperature (60, 80 and 100 °C) on water extraction of amaranth leaves has also been studied [14]. A significant effect of the solvent type, the method of extraction and their interaction on the content of total phenolic in the leaf extracts of *A. hypochondriacus* was found [48]. It states that the difference in total phenolic found between the studied extraction methods could be by the solvent temperature. It has also been stated that ethanol is a good solvent for chlorophyll extraction, and therefore the content of total phenolic may be related to the chlorophyll extracted. Unfortunately, there is no uniform methodology for extracting substances from plant material, nor is there a uniform organic solvent; what is chosen is often according to what is available. There is similar disunity in the methods of antioxidant determination of plant extracts. If we have to evaluate the antioxidant activity against the chosen radicals, it would be appropriate to use a verified generator and a suitable monitoring substance. General techniques for determining so-called redox potential, such as FRAP and DPPH, tell us about the ability to capture an electron at a pH that is only exceptionally in the human cells (FRAP is determined at pH 3.5–4.0 while DPPH at pH 7.0–7.4). However, the hydroxyl and nitritoperoxyl radicals react in a different way to the superoxide radical by incorporating the radical into the antioxidant molecule.

## 5. Conclusions

Total phenolic, flavonoid and rutin content, antioxidant activity against superoxide and hydroxyl radicals, the FRAP assay and DPPH activity were determined for ethanol extracts of dried leaves of the new Slovak amaranth varieties ‘Pribina’ and ‘Zobor’. We found higher antioxidant effects of the ethanol extract of the leaves of the amaranth ‘Pribina‘ variety compared to the extract of the ‘Zobor’ variety in all methods. The differences found were statistically significant. To evaluate the antioxidant activity and content of total phenolic, total flavonoids and rutin, the measured values were recalculated to the amount of dry matter because they take into account the content of extracted substance (dry matter) and the amount of phenolic substances in it. The antioxidant activity for all types of methods was influenced by the content of total phenolic substances, total flavonoids and rutin, which was confirmed by the high correlation coefficients. Phenolic, including flavonoids, are a complex of biologically active substances with a strong influence on antioxidant activity. High activity was also found with flavonoid rutin. In conclusion, with the use of different amaranth species, different plant materials, different extraction methods and different antioxidant assays, contradictory results are often obtained. The new varieties of amaranth, ‘Pribina‘and ‘Zobor’, appear to be a promising source of plant metabolites with antioxidant and antiradical activities.

## Figures and Tables

**Figure 1 antioxidants-10-01705-f001:**
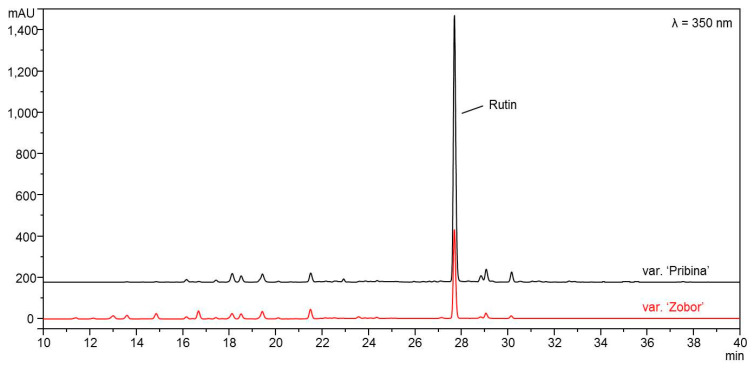
HPLC-DAD chromatograms of ethanol leaf extracts of two new Slovak amaranth varieties, *Amaranthus cruentus* L. variety ‘Pribina’ and *Amaranthus hypochondriacus* × *Amaranthus hybridus* L. variety ‘Zobor’.

**Table 1 antioxidants-10-01705-t001:** Dry matter, antioxidant activity of ethanol leaf extracts against hydroxyl and superoxide radicals, FRAP and DPPH activity of two amaranth varieties—‘Pribina’ and ‘Zobor’.

Parameter	‘Pribina’	‘Zobor’	*p* Value
DM [g.L^−1^]	12.65 ± 0.27 ^a^	11.27 ± 0.04 ^b^	*p* = 0.034
Superoxide [%.g^−1^ DM]	1.63 ± 0.14 ^a^	1.08 ± 0.09 ^b^	*p* = 0.008
Hydroxyl [%.g^−1^ DM]	3.20 ± 0.09 ^a^	2.13 ± 0.14 ^b^	*p* < 0.001
FRAP [µmoL.L^−1^.g^−1^ DM]	286.5 ± 11.7 ^a^	196.6 ± 8.9 ^b^	*p* < 0.001
DPPH IC50 [µg.mL^−1^ DM]	54.2 ± 1.78 ^a^	66.9 ± 0.74 ^b^	*p* < 0.001

Data represent the mean ± s.d. (standard deviation); a, b values showed differences between extracts; concentration of reference were: GAE 1 mmol.L^−1^; Superoxide SA 1 mmol.L^−1^ and in FRAP 0.1 mmol.L^−1^ GAE; GAE = Gallic acid equivalents; SA = sodium salicylate; DM = Dry matter.

**Table 2 antioxidants-10-01705-t002:** Content of total phenolic, flavonoids and rutin in ethanol leaf extracts of two amaranth varieties—‘Pribina’ and ‘Zobor’.

Parameter	‘Pribina’	‘Zobor’	*p* Value
Phenols [GAE mg.g^−1^ DM]	38.3 ± 1.01 ^a^	26.1 ± 0.33 ^b^	*p* < 0.001
Flavonoids [QE mg.g^−1^ DM]	26.5 ± 0.07 ^a^	20.3 ± 0.26 ^b^	*p* < 0.001
Rutin [mg.g^−1^ DM]	50.8 ± 0.24 ^a^	15.2 ± 0.05 ^b^	*p* < 0.001

Data represent the mean ± s.d. (standard deviation); a, b values showed differences between extracts; GAE = Gallic acid equivalents; DM = dry matter; QE = Quercetin equivalents.

**Table 3 antioxidants-10-01705-t003:** Correlation analysis.

Antioxidant Activity	Total Phenols	Total Flavonoids	Rutin
Superoxide anion radical scavenging activity	r = 0.935	r = 0.941	r = 0.938
Hydroxyl radical scavenging activity	r = 0.975	r = 0.973	r = 0.982
FRAP (ferric reducing ability of plasma) assay	r = 0.981	r = 0.980	r = 0.981
DPPH (2,2 Diphenyl 1 picrylhydrazyl)	r = 0.984	r = 0.983	r = 0.985

## Data Availability

The data presented in this study are available in article.

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
