# Peer review of "New Mutant Amaranth Varieties as a Potential Source of Biologically Active Substances"

_antioxidants, 2021, doi:10.3390/antiox10111705_

Round 1
Reviewer 1 Report
In the manuscript, the authors assessed the antioxidant activity by various methods, and also determined the content of TPC and TFC. The rutin content was determined by HPLC. The raw material for the research were new varieties of Slovak amaranth: 'Pribina' and 'Zobor'. These varieties have been the subject of research in the last few years.
Minor comments:
- Line 86. Did the authors only maceration of the raw material once? In the results, please indicate the extraction efficiency.
- Please also provide the determination values of the reference compounds in Table 1 In the tables and in the text, I suggest that you leave only one unit of conversion to dry weight (DM)
- The authors performed the determination of routine using the HPLC method. In the additional materials or the main text, I propose to present the HPLC chromatogram.
- In the available literature, I have not found detailed information on the comparison of both varieties in the parameters tested.
I recommend for publication
Author Response
Dear Reviewer.
Thank you very much for your time to review our article and help to improve it.
We tried to answer to the questions as well as corrected requested.
Rev1
In the manuscript, the authors assessed the antioxidant activity by various methods, and also determined the content of TPC and TFC. The rutin content was determined by HPLC. The raw material for the research were new varieties of Slovak amaranth: 'Pribina' and 'Zobor'. These varieties have been the subject of research in the last few years.
Minor comments:
- Line 86. Did the authors only maceration of the raw material once? In the results, please indicate the extraction efficiency.
- (authors) Maceration was provided only once. Although there were the same conditions for material preparation – drying, mortaring and maceration – there was noticed different effectiveness of extraction/maceration.
- Please also provide the determination values of the reference compounds in Table 1 In the tables and in the text, I suggest that you leave only one unit of conversion to dry weight (DM)
- (authors) Antioxidant activity of refferences are usually explained on the base of their amount of substances or concentrations. We have added infoaout concentrations of GAE and Superoxide SA under the Tab.1 and in the text
- The authors performed the determination of routine using the HPLC method. In the additional materials or the main text, I propose to present the HPLC chromatogram.
- (authors) We added chromatogram into the article text.
- In the available literature, I have not found detailed information on the comparison of both varieties in the parameters tested.
- (authors) Both tested varieties are very new and there was provided testing on them for the first time, so the presented article is original and there is no other publication for comparison.
Reviewer 2 Report
The title entitled “New mutant amaranth varieties as a potential source of biologically active substances” by Jozef Fejér , Ivan Kron , Adriana Eliašová , Daniela Gruľová * , Alena Gajdošová , Veronika Lancíková , Andrea Hricová reported that total phenolic, total flavonoid and rutin content, antioxidant activity against superoxide and hydroxyl radicals, FRAP (Ferric reducing ability of plasma) assay and DPPH (2,2-Diphenyl-1-picrylhydrazyl) radical scavenging assay were determined in ethanol extracts of dried leaves of new Slovak amaranth varieties ‘Pribina’ and ‘Zobor.
While I am familiar with the technique of measurement of antioxidant, I am not an expert in food varieties and cannot adequately comment on the amaranth.
1, The Fenton reaction is well known as a reaction in which iron and hydrogen peroxide generate hydroxyl radicals that decompose a variety of substances, but copper is also known to cause a similar reaction.
CUPRAC is also known as a test for antioxidant ability using copper.
J. Agric. Food Chem. 2004, 52, 7970−7981
2, The authors studied the following; Total phenolic, total flavonoid and rutin content, antioxidant activity against superoxide and hydroxyl radicals, FRAP assay and DPPH (2,2-Diphenyl-1-picrylhydrazyl) radical scavenging assay were 16 determined in ethanol extracts of dried leaves of new Slovak amaranth varieties ‘Pribina’ and ‘Zobor’.
The description of whether amaranth itself was originally high in antioxidant power, whether its antioxidant power did not decrease with breeding, or whether its antioxidant power increased with the new breed seemed vague.
I believe the paper will be of interest to the readership of antioxidant and would recommend it for acceptance after the minor points and annotated on the manuscript are addressed.
Author Response
Dear Reviewer.
Thank you very much for your time to review and help to improve our article.
We tried to respond our best.
Rev2
The title entitled “New mutant amaranth varieties as a potential source of biologically active substances” by Jozef Fejér , Ivan Kron , Adriana Eliašová , Daniela Gruľová * , Alena Gajdošová , Veronika Lancíková , Andrea Hricová reported that total phenolic, total flavonoid and rutin content, antioxidant activity against superoxide and hydroxyl radicals, FRAP (Ferric reducing ability of plasma) assay and DPPH (2,2-Diphenyl-1-picrylhydrazyl) radical scavenging assay were determined in ethanol extracts of dried leaves of new Slovak amaranth varieties ‘Pribina’ and ‘Zobor.
While I am familiar with the technique of measurement of antioxidant, I am not an expert in food varieties and cannot adequately comment on the amaranth.
1. The Fenton reaction is well known as a reaction in which iron and hydrogen peroxide generate hydroxyl radicals that decompose a variety of substances, but copper is also known to cause a similar reaction.CUPRAC is also known as a test for antioxidant ability using copper. J. Agric. Food Chem. 2004, 52, 7970−7981
(Authors) yes, Fenton reaction works also with copper and with other metals. We used standard method asi s described in M&M.
2. The authors studied the following; Total phenolic, total flavonoid and rutin content, antioxidant activity against superoxide and hydroxyl radicals, FRAP assay and DPPH (2,2-Diphenyl-1-picrylhydrazyl) radical scavenging assay were 16 determined in ethanol extracts of dried leaves of new Slovak amaranth varieties ‘Pribina’ and ‘Zobor’.
The description of whether amaranth itself was originally high in antioxidant power, whether its antioxidant power did not decrease with breeding, or whether its antioxidant power increased with the new breed seemed vague.
(Authors) In the present article, there were no comparison of new varietis with the those from which were bred (A. cruentus genotyp FICHA a hybrid K-433). Reached results were compared with the available publication, where was described antioxidant activity of Amaranthus cruentus and other Amaranth spp.